# Economic Analysis of Border Control Policies during COVID-19 Pandemic: A Modelling Study to Inform Cross-Border Travel Policy between Singapore and Thailand

**DOI:** 10.3390/ijerph20054011

**Published:** 2023-02-23

**Authors:** Celestine Grace Xueting Cai, Nigel Wei-Han Lim, Vinh Anh Huynh, Aparna Ananthakrishnan, Saudamini Vishwanath Dabak, Borame Sue Lee Dickens, Dian Faradiba, Sarin KC, Alec Morton, Minah Park, Chayapat Rachatan, Manit Sittimart, Hwee-Lin Wee, Jing Lou, Yot Teerawattananon

**Affiliations:** 1Saw Swee Hock School of Public Health, National University of Singapore (NUS), 12 Science Drive 2, #10-01, Singapore 117549, Singapore; 2Health Intervention and Technology Assessment Program (HITAP), Department of Health, Ministry of Public Health, 6th Floor, 6th Building, Tiwanon Road, Nonthaburi 11000, Thailand; 3Department of Management Science, University of Strathclyde, 16 Richmond Street, Glasgow G1 1XQ, UK; 4Department of Pharmacy, Faculty of Science, NUS, 18 Science Drive 4, Singapore 117559, Singapore

**Keywords:** economic evaluation, transmission, willingness to travel, COVID-19, border-opening policy

## Abstract

With countries progressing towards high COVID-19 vaccination rates, strategies for border reopening are required. This study focuses on Thailand and Singapore, two countries that share significant tourism visitation, to illustrate a framework for optimizing COVID-19 testing and quarantine policies for bilateral travel with a focus on economic recovery. The timeframe is the month of October 2021, when Thailand and Singapore were preparing to reopen borders for bilateral travel. This study was conducted to provide evidence for the border reopening policy decisions. Incremental net benefit (INB) compared to the pre-opening period was quantified through a willingness-to-travel model, a micro-simulation COVID-19 transmission model and an economic model accounting for medical and non-medical costs/benefits. Multiple testing and quarantine policies were examined, and Pareto optimal (PO) policies and the most influential components were identified. The highest possible INB for Thailand is US $125.94 million, under a PO policy with no quarantine but with antigen rapid tests (ARTs) pre-departure and upon arrival to enter both countries. The highest possible INB for Singapore is US $29.78 million, under another PO policy with no quarantine on both sides, no testing to enter Thailand, and ARTs pre-departure and upon arrival to enter Singapore. Tourism receipts and costs/profits of testing and quarantine have greater economic impacts than that from COVID-19 transmission. Provided healthcare systems have sufficient capacity, great economic benefits can be gained for both countries by relaxing border control measures.

## 1. Introduction

Tourism contributes significantly to the global economy, especially among Southeast Asian countries, including Singapore (SG) and Thailand (TH). Prior to the coronavirus disease 2019 (COVID-19) pandemic, the number of arrivals for TH was 39.9 million in 2019, which contributed US $64.4 billion or 12% of the nominal gross domestic product (GDP) [1]. SG received 19.12 million tourists in the same year, which contributed US $20.7 billion (S $27.7 billion) or 5.5% of nominal GDP [2].

Bilateral tourism between TH and SG is significant. In 2019, there were a total of 528,547 travelers from TH travelling to SG and 1,113,067 travelers from SG visiting TH [2,3]. However, with the arrival of COVID-19 in late 2019 and border closures and restrictions from March 2020 to suppress the importation of COVID-19 cases, tourism for both TH and SG was significantly reduced. Due to waves of restrictions, domestic lockdowns and border closures, TH lost 76% of its tourism receipts (US $15.4 billion in 2020) and SG lost 83% (US $3.7 billion in 2020) in total [4]. This loss of tourism devastated TH’s economy due to TH’s high dependence on the tourism sector.

Since 2021, the two governments have been transitioning to an endemic COVID-19 strategy by vaccinating their respective populations against COVID-19 and planning for the relaxation of border and pandemic control measures. Border control measures, including the quarantine and repeated testing of travelers, help to limit the number of cases imported into the local community but also increase the monetary and opportunity costs incoming travelers must bear. As part of the transition, border control measures will be relaxed, and evidence is required to support decisions on the appropriate degree of border control.

We therefore conducted this economic analysis, which aims to optimize COVID-19 testing and quarantine policies between TH and SG with a focus on economic recovery. A cost–benefit analysis (CBA) was adopted to allow comparison across multiple policy options. Since an increase in travel cost will lead to a fall in travel demand [5], we modelled the trade-off between the societal cost of imported and secondary cases in local communities, and the societal economic benefits of tourism receipts. The costs and profits of implementing testing and quarantine policies were also quantified. Case counts were modelled based on a previous analysis [6], adding on vaccination effectiveness and adjusting for travelers’ itineraries from one-way to round-trips.

## 2. Materials and Methods

This is a model-based study using epidemiological and economic modelling techniques to predict the impacts of bilateral testing and quarantine policy choices on economic outcomes and COVID-19 transmission during the COVID-19 pandemic. For a given bilateral testing and quarantine policy, the economic losses and gains of each country were quantified through three sub-models (Figure 1). First, a willingness-to-travel (WTT) model predicted the number of travelers from each country based on the given bilateral policy. Second, the transmission model estimated the number of these travelers who would be detected with COVID-19 at each stage of travel, the number of secondary cases caused in the community by infectious travelers missed at the border, and the probability of symptoms in each group. Lastly, an economic model calculated the net monetary benefit (NMB) of each country using the numbers of travelers and COVID-19 cases, taking into account tourism receipts, aviation receipts, cost/profit from testing and quarantine, reduced local expenditure due to travel, and cost and health loss due to COVID-19 cases. The values and data sources of model parameters are available in Appendix A.

### 2.1. Study Population, Setting and Time Horizon

The study population consists of the population of SG and TH. The timeframe of this study is the month of October 2021, when both countries began to relax cross-border measures. This study assumes that only vaccinated people are allowed to travel, as a relatively safe step to boost the economy. At that time, the COVID-19 prevalence was 0.59% in SG [7] and 0.14% in TH [8]. By October 2021, 84% of the Singaporean population had been vaccinated [9] with either Pfizer (BNT162b2 [10]/Comirnaty [11]) or Moderna (Spikevax [12]), and 35% of the Thai population [13] with either AstraZeneca (Vaxzevria [14]) or Sinovac (CronaVac [15]). Although the time horizon was one month, a life-time horizon was applied when quantifying the health and productivity loss from COVID-19 mortality.

### 2.2. Comparators

We examined nine testing strategies (Table 1), each with a possible quarantine length of 0 to 14 days, applied independently to TH and SG for a total of 16,384 different policy combinations. Only an antigen rapid test (ART) was conducted during quarantine, whilst pre-testing, entry testing and exit testing could involve either an ART or polymerase chain reaction test (PCR). Testing frequency during quarantine could be daily (Strategies S1 and S5), every 3 days (Strategies S2 and S6), weekly (Strategies S3 and S7), or absent (Strategy S9). PCR is assumed to have a turnaround time of 1 day. Thus, when PCR was used as an entry test (Strategies S5–S8), the quarantine length was at least 1 day.

### 2.3. Willingness-to-Travel Model

The WTT model aimed to predict the monthly number of travelers for the different bilateral strategies. The WTT model was trained and tested using historical monthly data from June 2015 to June 2021, covering the time period before and during both countries’ COVID-19 travel quarantine measures. Historical data on the number of travelers between SG and TH were provided by Singapore Tourism Board [16] and Immigration Bureau Thailand [3], respectively. We built separate models for tourists and business travelers, because tourists may be more sensitive to change in quarantine length than business travelers.

The data were fit with different predictors (length of quarantine, testing policy, prevalence) and functional forms (polynomial, exponential, summation or separation of quarantine length in origin and destination countries). Model selection was based on predictive accuracy (measured by mean absolute predictive error (MAPE)) and interpretability. An exponential model with sum of quarantine lengths as the predictor was chosen for its relatively low MAPE and good interpretability. This study employed the following exponential WTT model (Demand is the monthly number of travelers; Qlength is the sum of quarantine lengths in the origin country and destination country):(1)$$Demand_{SG\to TH,\;tourist}=65,253\times {\left(1-22.3\%\right)}^{Qlength}$$
(2)$$Demand_{TH\to SG,\;tourist}=39,514\times {\left(1-22.3\%\right)}^{Qlength}$$
(3)$$Demand_{SG\to TH,\;biz}=2632\times {\left(1-12.6\%\right)}^{Qlength}$$
(4)$$Demand_{TH\to SG,\;biz}=4818\times {\left(1-12.6\%\right)}^{Qlength}$$

Based on the equations above, without quarantine, there would be 65,253 tourists from SG to TH, 39,514 tourists from TH to SG, 2632 business travelers from SG to TH, and 4818 business travelers from TH to SG per month. When quarantine length increases by 1 day, the number of tourists from both sides decreases by 22.3%, and the number of business travelers from both sides decreases by 12.6%. As we assumed only vaccinated people are allowed to travel, the predicted number of travelers from each country is multiplied by the vaccination coverage rate of the country.

### 2.4. Transmission Model

The transmission model aimed to predict the number of cases diagnosed at each stage of travel and the number of secondary cases caused by travelers in the community. The transmission model used Monte Carlo simulation to estimate testing, quarantine and COVID-19 transmission outcomes for travelers from country A (origin country, either SG or TH) to country B (destination country, either SG or TH). COVID-19 prevalence was assumed as 0.59% in SG (7) and 0.14% in TH (8) and the reproduction rate of COVID-19 (R_0_) was assumed as 7, a high estimate for the COVID-19 Delta variant, to be conservative. When exploring the various policies of testing and quarantine (Table 1), the model simulated a cohort of travelers initially infected in country A as well as a cohort of travelers only infected during their stay in country B (Figure 2).

For the cohort infected in country A, the travelers undergo the quarantine and testing policies imposed by country B on incoming travelers, and travelers who yield positive testing results or symptoms before the end of quarantine are removed (tabulated as diagnosed by pre-test, during quarantine or by exit-test at country B). The travelers then exit quarantine and cause transmission during their stay in country B, until they recover from their infection or they leave for country A. They then undergo the quarantine and testing policies imposed by country A on returning travelers, and travelers who yield positive testing results or symptoms before the end of quarantine are removed in the same manner as in country B. Finally, the travelers exit quarantine and cause transmission in country A until they fully recover.

For the cohort infected at country B, only the quarantine and testing policies imposed by country A on returning travelers and the post-quarantine transmission in country A were simulated, as these travelers are uninfected prior to the completion of testing and quarantine in country B. However, the transmissions caused in country B by these travelers after being infected before they leave country B were included.

Transmission caused by each traveler was computed as the ratio of the number of days spent in a given country while infected (post-incubation period) to the mean infectious period, multiplied by the assumed value of R_0_ in the country. More details on the transmission model are provided in Appendix A.

### 2.5. Economic Model

A societal perspective was adopted in this study, as the impacts of COVID-19 extend beyond the healthcare sector. Economic and health gains and losses were discounted by 3% per year for health and productivity losses due to premature mortality, as recommended by the economic evaluation guidelines in both countries [17,18]. The currency used in this study was the 2021 US dollar (USD). Price data were collected for the period of June through October 2021 from publicly available data sources, advice from local government agencies and estimates (available in Appendix A). No conversion was needed to adjust for differences between price levels over time.

The economic model quantified the NMB of each country as the sum of the 13 components described in Table 2. The costs and benefits were classified into four groups. The first cost/benefit group included traveler-related economic costs and benefits. The tourism sectors earn tourism receipts from incoming travelers as well as aviation receipts from travelers who visit their destination country. When people travel to a country, their expenditure in the home country’s market is reduced. To measure such opportunity costs of allowing international travel, we quantified the local expenditure of residents who choose not to travel. For both sides, travelers’ expenditure on quarantine is considered to be a net economic gain to the destination country because empty hotels are a sunk cost and hotel staff may lose employment should there be no travelers. The cost of conducting ART was estimated to be US $7.5 in SG and US $10.0 in TH. Unlike ART, PCR testing capacity was more limited in both countries, so it was assumed that PCR for travelers is mainly offered by private healthcare providers, with the total cost of conducting PCR estimated to be US $46.90–US $60.0 in SG and US $34.75 in TH. Based on those assumptions, we quantified the profit from implementing a bilateral policy on incoming travelers, and the cost of implementing a bilateral policy on travelers when they travel to their destination and return.

The second cost/benefit group included the treatment costs of COVID-19 cases. It was assumed that when a traveler was diagnosed in their destination country, all medical costs were paid by the travelers themselves. Therefore, we included the treatment cost of travelers who are diagnosed with COVID-19 at the border or in the community of either country. The secondary community cases caused by incoming travelers and returning travelers who are not filtered out by the testing and quarantine policy at the border, also incur treatment costs to a country. Moreover, people who choose not to travel may contribute to local transmission. Hence we also added the treatment cost of community cases from local transmission.

The third cost/benefit group included non-medical costs related to COVID-19 transmission. This included the cost of test-trace-isolation (TTI) for close contacts of detected cases in the community, productivity loss due to COVID-19 morbidity and mortality, and productivity loss from the quarantining of close contacts. We assumed 30% productivity loss if a person is quarantined with no COVID-19 symptoms [19], and 100% productivity loss if a person is having mild/moderate/severe COVID-19 symptoms or requires an ICU hospitalization stay.

The fourth cost/benefit group included health losses due to quarantine and COVID-19 morbidity and mortality. The demographic and risk profiles of travelers and the general population were assumed to be exogenously dependent on the age structure of the population group. The quality-adjusted life-year (QALY) loss due to morbidity for unvaccinated patients comes from US patient estimates [20]. We assumed that the QALY loss due to morbidity for vaccinated patients was 90% that of unvaccinated patients at the same disease severity because vaccinated patients recover more quickly than unvaccinated peers of the same age group. After being quantified in terms of QALY, health losses were expressed in monetary terms through multiplication with willingness-to-pay, which was assumed to be 1 GDP per capita of the country in the base case analysis.

### 2.6. Measurement and Valuation of Outcomes

The size of gain under each bilateral border-opening policy P is measured by the incremental net benefit (INB) of each country compared to the pre-opening policy $P^{*}$, i.e.,
(5)$$INB_{TH}\left(P\right)=NMB_{TH}\left(P\right)-NMB_{TH}\left(P^{*}\right)$$
(6)$$INB_{SG}\left(P\right)=NMB_{SG}\left(P\right)-NMB_{SG}\left(P^{*}\right)$$

The reference bilateral policy $P^{*}$ for SG included a 14-day quarantine, PCRs pre-departure, upon arrival, at the end of quarantine and a weekly ART during quarantine (Strategy S7 in Table 1). $P^{*}$ for TH included a 14-day quarantine, PCRs pre-departure, upon arrival and at the end of quarantine, with no routine testing during the quarantine period (Strategy S8 in Table 1).

As the objective of this study is to optimize COVID-19 testing and quarantine policies between TH and SG, an ideal optimal bilateral policy should maximize the INB of SG and TH simultaneously. However, two countries may not reach their highest possible INB simultaneously under one bilateral policy due to trade-offs. Therefore, we examined the possibility of a Pareto optimal (PO) bilateral policy wherein there is no other bilateral policy that can further increase one country’s INB without lowering the other country’s INB.

According to policymakers in SG and TH, the net economic gain may not be the only criterion for policy choice. Transmission numbers, deaths, and intensive care unit (ICU) occupancy are also informative for decision making. Therefore, we tabulated the transmission results in each country for each PO policy. Imported cases include (i) inbound travelers detected at the border and (ii) outbound travelers detected upon return. Secondary cases include community cases infected by inbound travelers or returning outbound travelers who are not successfully picked up by the testing and quarantine policy at the border. On top of that, deaths and the number of ICU cases are tabulated for incoming travelers, returning travelers, secondary cases from incoming travelers, and secondary cases from returning travelers. The numbers are tabulated separately since they may affect policy making differently.

### 2.7. Deterministic Sensitivity Analysis (DSA)

To test the sensitivity of findings in the base case analysis, we first increased the daily infection rate and prevalence in both countries by 50%. Secondly, R_0_ was lifted to 10 as a high estimate for the COVID-19 Omicron variant which became prevalent around December 2021 [21]. Thirdly, a higher vaccination coverage rate in both countries (92% for SG [22] and 76% for TH [23] as in January 2023) was explored, with a bigger group size of vaccinated travelers and better protection from vaccination in the community.

Fourthly, the percentage reduction in number of travelers with one more day of quarantine was increased and reduced by 50% for Singaporean/Thai tourists and business travelers simultaneously. Fifthly, the base case analysis did not take into account the spillover effect of the tourism sector to be conservative about the gain from tourism; in the DSA, this spillover effect was quantified using a tourism multiplier of 2.35 for Singapore (derived based on an input-output table in 2017 [24] and historical tourism receipt components of SG from Thai visitors during the period 2016–2020 [16]) and 2.09 for Thailand [25]. Sixthly, we explored another scenario in which vaccinated and unvaccinated individuals are allowed to travel under the same testing and quarantine policy.

Seventhly, the medical costs of COVID-19 cases were doubled for both SG and TH. Eighthly, the productivity loss due to quarantine/isolation was reduced from 30% (base case value) to 0%, which is possible with the maturity of remote working. Ninthly, the assumption that vaccination provided a 10% reduction in QALY loss for symptomatic cases was changed to a 0% reduction. Lastly, the willingness-to-pay for 1 QALY was assumed to be 1 GDP per capita of each country in the base case analysis, and we checked the high estimate of 3 GDP per capita in the DSA [26].

## 3. Results

We estimated the INB of both TH and SG under various combinations of testing and quarantine bilateral policies (Figure 3). There are three PO policies and all do not quarantine at either border. One PO policy has testing strategies where both TH and SG implement ART pre-departure and upon arrival (Strategy S4 in Table 1). This gives TH a monthly INB of US $125.94 m (highest INB for TH across all policies) and SG US $27.36 m. Another PO policy has testing strategies where TH does not require any test on travelers (Strategy S9 in Table 1) whilst SG requires ART pre-departure and upon arrival (Strategy S4 in Table 1). This gives TH a monthly INB of US $123.97 m and SG US $29.78 m (highest INB for SG across all policies). One last PO policy has testing strategies where SG does not require any test on travelers (Strategy S9 in Table 1) whilst TH requires ART pre-departure and upon arrival (Strategy S4 in Table 1). This gives TH a monthly INB of US $125.14 m and SG US $27.78 m.

The bilateral policy when TH has the lowest INB (Figure 3) is when TH requires a 14-day quarantine with no testing (Strategy S9 in Table 1), and SG requires a 14-day quarantine with PCRs pre-departure, upon arrival, and quarantine exit, with daily ARTs during quarantine (Strategy S5 in Table 1). This gives TH a negative monthly INB of -US $0.02 m. For SG, the lowest INB is when SG requires no quarantine with ART pre-departure and upon arrival (Strategy S8 in Table 1), and TH requires a 5-day quarantine with PCRs pre-departure, upon arrival, and quarantine exit, with daily ARTs during quarantine (Strategy S5 in Table 1). This gives SG a negative monthly INB of −US $10.36 m. The lowest combined INB is when TH and SG both require a 14-day quarantine and PCRs pre-departure, upon arrival, and quarantine exit, with daily ARTs during quarantine (Strategy S5 in Table 1). This gives TH a monthly INB of US $0.03 m and SG a monthly INB of −US $0.06 m, for a total of −US $0.03 m per month.

We calculated the number of cases in each country under each PO policy, including the number of imported cases, secondary cases, ICU cases and deaths from incoming travelers, returning travelers, secondary cases from incoming travelers and secondary cases from returning travelers (Table 3). Regardless of the PO policy, the total number of ICU cases were below nine people per month for TH and below two people per month for SG, and the total number of deaths were below four people per month for TH and below one person per month for SG. This indicates that the PO policies are feasible in terms of disease control. Requiring testing pre-departure and upon arrival greatly reduces the number of imported and secondary cases in a country. Most infectious travelers are detected by the pre-departure test and are prevented from leaving the country of origin.

To determine drivers of costs and benefits, we studied the range of the 13 cost/benefit components across all bilateral policies, where a larger range indicated that the cost/benefit component is more influential to the INB of the country. For TH, the five most influential components come from tourism receipts from travelers coming from SG, as well as the cost and profit of implementing testing and quarantine policies (Figure 4A). For SG, these are the cost of testing when SG travelers go to TH and return, tourism receipts from TH travelers, reduced local expenditure when SG travelers go to TH, the cost of quarantine when SG travelers go to TH and return, and aviation receipts from TH travelers (Figure 4B). For both countries, tourism receipts and the cost/profit of testing and quarantine policies have a bigger economic impact than COVID-19 transmission.

The no-quarantine characteristic of PO policy is robust to alternative model assumptions in the DSA (Table 4). Regardless of DSA scenario, the testing strategies where SG requires ART pre-departure and upon arrival (Strategy S4 in Table 1) whilst TH does not require any test on travelers (Strategy S9 in Table 1) or requires ART pre-departure and upon arrival (Strategy S4 in Table 1) always remain PO. Nevertheless, the testing strategy where SG does not require any test of travelers (Strategy S9 in Table 1) whilst TH requires ART pre-departure and upon arrival (Strategy S4 in Table 1) is not PO when the daily infection rate increases by 50%, when R_0_ is 10, and when willingness-to-pay is 3 GDP per capita. The INB per month under PO policies is generally robust. A significant increment in INB is observed when the spillover effect of tourism sector is taken into account, when vaccination coverage expands (more individuals are allowed to travel), and when unvaccinated individuals are allowed to travel under the same policy as vaccinated travelers. The expansion of vaccination coverage increases INB without bringing further ICU burden to the healthcare system. In contrast, the increase in INB from allowing vaccinated and unvaccinated individuals to travel under the same PO policies is associated with surging critical case counts, which needs further evaluation by decision makers on whether the critical care demands could be tolerated by both countries’ healthcare systems.

## 4. Discussion

This study aims to optimize COVID-19 testing and quarantine policies between TH and SG with a focus on economic recovery. All the PO policies have no quarantine at both borders because tourism receipts are an influential cost/benefit component. As a longer quarantine period reduces people’s willingness to travel, we observe a large shrinkage in tourism receipts, thus cutting economic gains substantially. Balancing economic losses and protection of national healthcare systems has been of utmost priority to policymakers in both SG and TH. Both countries, however, recognize the need to open borders to support millions of livelihoods and economic recovery [20,27].

To contextualize this study, we found that the governments of both TH and SG had been largely in favor of more open and relaxed travel policies between the two countries. From November 2021 [28], vaccinated travelers from SG were able to apply for entry to TH via the online “Thailand Pass” system which exempted the previous 14-day quarantine and only required a pre-departure test and an entry test (Test and Go scheme) for fully vaccinated travelers. From April and May 2022 onwards, Thai policymakers removed the need for pre-departure and on-arrival testing due to the strengthening of the healthcare system [29,30]. Similarly, in December 2021, SG implemented the vaccinated travel lane (VTL) which allowed quarantine-free travel from TH to SG [31], followed by a gradual relaxation of testing requirements in early 2022 [32,33,34], culminating in the complete removal of testing requirements from April 2022 onwards. These real-world policies are consistent with our study results, and demonstrated that both countries recognize the importance of quarantine-free and light-to-no-testing travel policies to enable economic recovery in the two countries. The timeframe of restriction relaxation has differed between the two countries, and can be attributed to differences in local health systems, with Singapore experiencing a higher rise in number of COVID-19 deaths in early 2022. More details about the countries’ respective travel policies and corresponding timeline can be found in Appendix A.

The COVID-19 Delta variant was modelled in the base case analysis to reflect the prevalent strain during the study timeframe of October 2021. In DSA, R_0_ was increased to 10 to model the more highly transmissible Omicron variant which became prevalent in December 2021. Since the traveler-related economic costs/benefits, including tourism receipts and the cost/profit of implementing border control policies, have a larger economic impact than COVID-19 transmission, the results for Omicron resemble the findings from Delta, except for fewer PO policy options and more detected imported cases and local transmission from undetected imported cases. However, as Omicron has a higher rate of being asymptomatic and a lower mortality rate, there may be a smaller change in ICU case numbers than estimated in DSA.

Existing studies explored the economic impacts of border control policies on aspects including the stock market, tourism industry and passenger and cargo transportation [35]. However, those studies did not compare the economic impact in non-health sectors and healthcare costs and productivity losses from COVID-19 transmissions resulting from cross-border travel under various testing and quarantine policy choices. This study utilizes the cost–benefit analysis method to quantify the net economic benefit of 16,384 testing and quarantine policies for travelers in a bilateral travel framework, combining the receipt revenue in the tourism and aviation sectors, the medical costs, health and productivity losses from COVID-19 transmissions resulting from cross-border travel, and costs and profits from testing and quarantine policy implementation.

The base case analysis assessed the impact of vaccinated travelers only as this was a relatively safe choice to boost the economy and monitor disease control at the time of the study in October 2021. In DSA, uniform border reopening policies for unvaccinated and vaccinated individuals were explored to address equity concerns. The model framework can be easily applied to evaluate alternative border reopening policy options allowing unvaccinated individuals to travel under different policies from vaccinated travelers.

This model framework can be generalized to any other pair of countries, by replacing model parameters with local data to inform policy decisions on bilateral border control. Data used in this model should be available in all countries, but accessibility might vary. For example, the monthly numbers of inbound travelers to SG are publicly available while the numbers to TH need to be requested from the TH government. It is important to note that there is asymmetry between SG and TH, as TH has a more tourism-dependent economy, while for SG there is a trade-off between pandemic control and economic recovery. This observation is expected for almost all country pairs and requires the cooperation of both origin and destination countries for decisions on appropriate border controls. Additionally, the benefits of opening borders apply to the entire population, whereas the costs of COVID may only apply to a smaller subset when outbreaks are waning.

Our study has several limitations. First, the MAPE of the current WTT models are around 40. This is mainly because the testing data sample (February–June 2021) has a very small number of travelers. The WTT model could be re-estimated to improve its predictive power when the tourism data of both countries from the border re-opening period becomes available. Second, the length of stay is assumed to be constant regardless of quarantine length, when in reality people who are willing to travel under long quarantine will stay longer. Thus for long-quarantine scenarios, both tourism receipts and transmission may be underestimated. However, according to the Singapore Tourism Board [16], inbound travelers pre-COVID and during-COVID have a similar amount of expenditure throughout their stay in SG, although the latter stay much longer. Furthermore, costs related to COVID-19 transmission have been found to have a small impact on INB. Therefore, our findings should be relatively robust with this assumption.

Third, intangible gains from overseas travel are not quantified in this study due to the unavailability of such data. Such intangible gains may include improved mental wellbeing of travelers from having a vacation overseas and its downstream positive effects on work productivity upon return, as well as emotional gains from visiting family members. When people residing in SG are willing to travel, such intangible gains should exceed all the costs borne by the travelers. Fourth, prevalence rates may change over time between countries as localized outbreaks may occur. Higher prevalence rates will require more stringent testing and quarantine policies, depending on whether local healthcare resources, such as ICU capacity, are sufficient. Lastly, additional cases beyond secondary cases are not accounted for and may result in an underestimation of the total number of cases and overestimation of the INB.

## 5. Conclusions

In conclusion, opening borders can bring significant economic benefits to both TH and SG using the PO bilateral policies of (i) no quarantine with ART pre-departure and upon arrival on either side, (ii) no quarantine on either side, no testing to enter SG, and ART pre-departure and upon arrival to enter TH, (iii) no quarantine on either side, no testing to enter TH, and ART pre-departure and upon arrival to enter SG. Provided the healthcare systems are both robust and prepared to receive potential imported cases and resulting secondary cases, travel between TH and SG could resume with a continual review of border control policies based on local COVID case counts.

## Figures and Tables

**Figure 1 ijerph-20-04011-f001:**
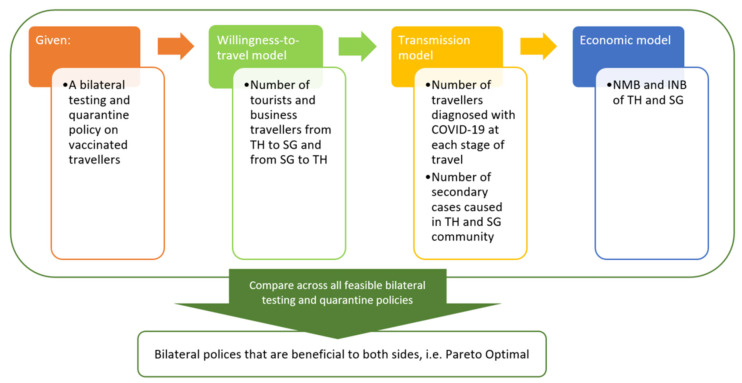
**Whole model diagram.** Abbreviations: COVID-19, coronavirus disease 2019; SG, Singapore; TH, Thailand; NMB, net monetary benefit; INB, incremental net benefit. The whole model diagram illustrates how sub-models work together to identify the Pareto optimal bilateral policies of testing and quarantine.

**Figure 2 ijerph-20-04011-f002:**
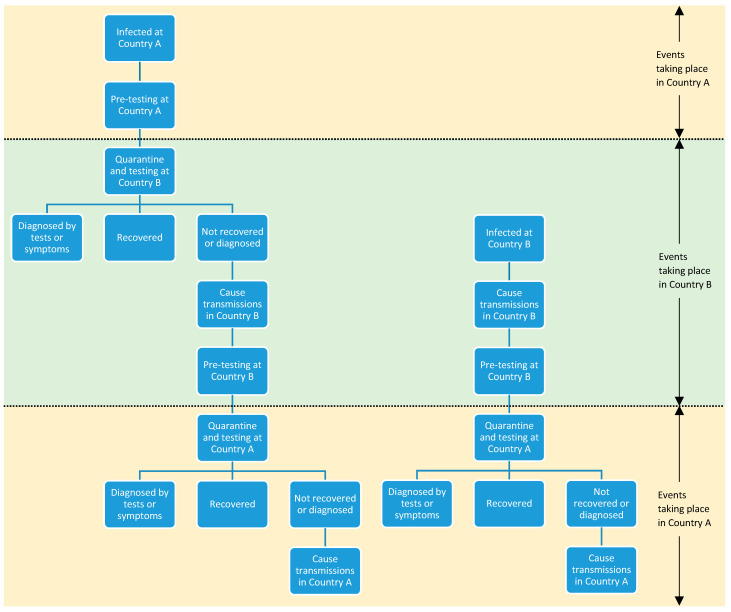
**Transmission diagram of a round trip from Country A to Country B.** The left diagram illustrates the possible itineraries of the cohort infected in Country A (home country) before they start the trip. The right diagram illustrates the possible itineraries of the cohort infected in Country B (destination country).

**Figure 3 ijerph-20-04011-f003:**
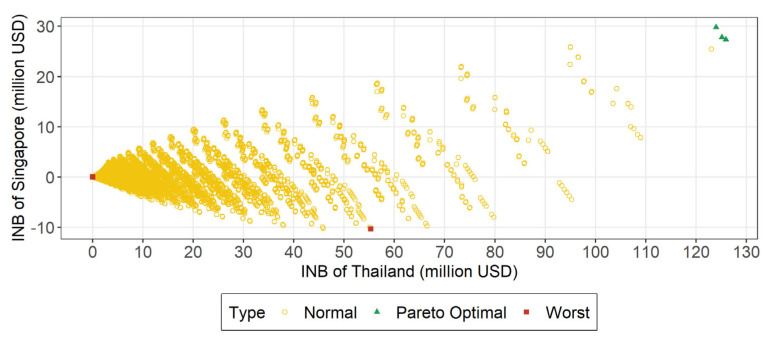
**Incremental net benefit of Singapore and Thailand under all bilateral policy options, with Pareto optimal policies and worst policies highlighted.** Abbreviations: SG, Singapore; TH, Thailand; INB, incremental net benefit; PO, Pareto optimal; m, million. Horizon: 1 month. Unit of INB for horizontal and vertical axes: million USD. Each hollow dot represents one bilateral policy option. The horizontal coordinate is the INB of TH and the vertical coordinate is the INB of SG. Three PO policy pairs are highlighted as triangle dots: (i) Both TH and SG do not require quarantine but require ARTs pre-departure and upon arrival to enter both TH and SG (S4), with INB of TH at US $125.94 m and SG at US $27.36 m; (ii) TH requires no quarantine and no testing (S9), whilst SG requires no quarantine but requires ARTs pre-departure and upon arrival to enter SG (S4), with INB of TH at US $123.97 m and SG at US $29.78 m; (iii) SG requires no quarantine and no testing (S9), whilst TH requires no quarantine but requires ARTs pre-departure and upon arrival to enter SG (S4), with INB of TH at US $125.14m and SG at US $27.78 m. Worst policy pairs are highlighted as square dots. Worst for TH: TH requires 14-day quarantine but no testing (S9), whilst SG requires 14-day quarantine, PCR pre-departure, upon arrival, and quarantine exit tests, with daily ART during quarantine (S5), giving TH an INB of −US $0.02m and SG US $0.02m. Worst for SG: TH requires 5-day quarantine, PCR pre-departure, upon arrival, and quarantine exit tests, with daily ART during quarantine (S5), whilst SG requires no quarantine but requires ARTs pre-departure and upon arrival (S4), giving TH an INB of US $55.31 m and SG −US $10.36 m. Worst for TH and SG combined (total INB): Both TH and SG require 14-day quarantine, PCR pre-departure, upon arrival, and quarantine exit tests, with daily ART during quarantine (S5), resulting in a total INB of −US $0.03 m, with INB of TH at US $0.03 m and of SG at −US $0.06 m.

**Figure 4 ijerph-20-04011-f004:**
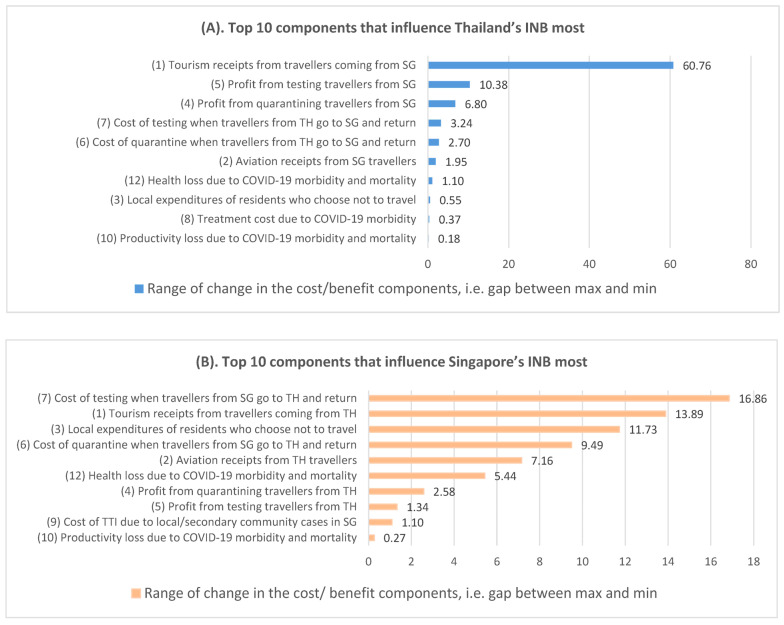
**Influential components of incremental net benefit.** Abbreviations: COVID-19, coronavirus disease 2019; SG, Singapore; TH, Thailand; INB, incremental net benefit; TTI, test-trace-isolation. Ten components with the largest range for TH are sorted in (**A**); ten components with the largest range for SG are sorted in (**B**). A larger range indicates a more influential cost/benefit component to the country’s INB. The numbering in front of each cost/benefit component corresponds to that in Table 2.

**Table 1 ijerph-20-04011-t001:** Policy choices examined in this study.

Strategy Number	Pre-Test	Entry Test	Exit Test	Test during Quarantine		
				Weekly	Every 3 Days	Daily
S1	ART	ART	ART			ART
S2	ART	ART	ART		ART	
S3	ART	ART	ART	ART		
S4	ART	ART	ART			
S5	PCR	PCR	PCR			ART
S6	PCR	PCR	PCR		ART	
S7	PCR	PCR	PCR	ART		
S8	PCR	PCR	PCR			
S9	Quarantine only					

Abbreviations: ART, antigen rapid test; PCR, polymerase chain reaction. The turnaround time of PCR is 1 day. Quarantine length is at least 1 day when implementing Strategies S5–S8.

**Table 2 ijerph-20-04011-t002:** Cost/benefit components from country A’s perspective.

**Traveler-related economic benefits/costs**	(+) Tourism receipts by travelers from country B (inclusive of aviation receipts) (+) Aviation receipts by travelers from country A visiting country B (+) Local expenditures of residents of country A who choose not to travel (+) Profit from quarantining travelers from country B (+) Profit from testing travelers from country B (−) Cost of quarantine when travelers from country A go to country B and return (−) Cost of testing when travelers from country A go to country B and return
**COVID-19-case-related medical costs**	8. (−) Treatment cost due to COVID-19 morbidity (−) Treatment cost of outgoing cases diagnosed by pre-test at country A’s border (−) Treatment cost of outgoing cases diagnosed at country B’s border (−) Treatment cost of outgoing cases missed by country B’s border but who develop symptoms afterwards (−) Treatment cost of returning cases diagnosed by pre-test at country B’s border (−) Treatment cost of returning cases diagnosed at country A’s border (−) Treatment cost of returning cases missed by country A’s border but who develop symptoms afterwards (−) Treatment cost of secondary community cases in country A, infected by travelers coming/returning from country B (−) Treatment cost of community cases from local transmission in country A
**COVID-19 cases-related non-medical costs**	9. (−) Cost of TTI due to local/secondary community cases in country A 10. (−) Productivity loss due to COVID-19 morbidity and mortality 11. (−) Productivity loss due to quarantine
**COVID-19 cases-related health loss**	12. (−) Health loss due to COVID-19 morbidity and mortality (QALY in monetary term) 13. (−) Health loss due to quarantine (QALY in monetary term)

Abbreviations: COVID-19, coronavirus disease 2019; TTI, test-trace-isolation; QALY, quality-adjusted life-year. The (+)/(−) sign in front of each item indicates whether the item is of positive or negative value. Country A could be either Singapore or Thailand, and country B will be the other.

**Table 3 ijerph-20-04011-t003:** Transmission results under Pareto optimal policies (all numbers are per-month values).

	TH Testing	SG Testing	TH Quarantine Length (Days)	SG Quarantine Length (Days)	Imported Cases	Secondary Cases	Inbound Travelers				Returning Outbound Travelers			
							ICU Cases among Travelers	ICU Cases among Secondary Cases	Deaths among Travelers	Deaths among Secondary Cases	ICU Cases among Travelers	ICU Cases among Secondary Cases	Deaths among Travelers	Deaths among Secondary Cases
TH	S4	S4	0	0	54.158	159.116	0.024	0.876	0.007	0.382	0.024	1.518	0.014	0.661
	S4	S9	0	0	28.152	161.187	0.010	0.878	0.003	0.382	0.022	1.548	0.015	0.674
	S9	S4	0	0	149.931	546.419	0.086	4.773	0.027	2.078	0.037	3.450	0.014	1.503
SG	S4	S4	0	0	23.231	18.274	0.014	0.011	0.005	0.005	0.131	0.170	0.048	0.073
	S4	S9	0	0	29.686	102.709	0.018	0.055	0.007	0.024	0.145	0.962	0.048	0.416
	S9	S4	0	0	22.996	40.661	0.001	0.011	0.000	0.005	0.101	0.392	0.058	0.169

Abbreviations: SG, Singapore; TH, Thailand. Testing strategy (when no quarantine): (S4) ART pre-departure and on arrival; (S9) no testing. Imported cases include (i) inbound travelers detected within the border and (ii) outbound travelers detected after return. Secondary cases include those infected by inbound travelers or returning outbound travelers.

**Table 4 ijerph-20-04011-t004:** Pareto optimal policies from Deterministic Sensitivity Analysis (all numbers are per-month value).

Condition	TH Strategy	SG Strategy	TH Quarantine Days	SG Quarantine Days	TH INB (Millions)	SG INB (Millions)	TH Critical Cases	SG Critical Cases	Total Cases	Total Deaths
Base Case	S4	S4	0	0	125.943	27.363	2.44	0.33	254.78	1.20
	S4	S9	0	0	125.139	27.782	2.46	1.18	321.73	1.57
	S9	S4	0	0	123.973	29.776	8.35	0.50	760.01	3.85
Daily infection rate (1.5×)	S4	S4	0	0	125.528	28.046	3.64	0.49	380.63	1.79
	S9	S4	0	0	123.178	30.427	12.50	0.75	1138.38	5.77
R_0_ (R_0_ = 10)	S4	S4	0	0	125.750	27.210	3.47	0.40	330.80	1.68
	S9	S4	0	0	123.311	29.436	11.87	0.68	1011.61	5.46
Vaccine coverage (SG: 92%, TH 76%)	S4	S9	0	0	140.156	59.513	2.13	1.15	492.92	1.42
	S4	S4	0	0	140.056	59.784	2.12	0.37	398.86	1.08
	S9	S4	0	0	138.747	62.375	6.31	0.47	1008.41	2.95
Percentage reduction in number of travelers with one more day of quarantine (0.5×)	S4	S4	0	0	116.373	29.659	2.44	0.33	254.78	1.20
	S4	S9	0	0	115.569	30.079	2.46	1.18	321.73	1.57
	S9	S4	0	0	114.403	32.073	8.35	0.50	760.01	3.85
Percentage reduction in number of travelers with one more day of quarantine (1.5×)	S4	S4	0	0	126.361	27.390	2.44	0.33	254.78	1.20
	S4	S9	0	0	125.557	27.809	2.46	1.18	321.73	1.57
	S9	S4	0	0	124.391	29.803	8.35	0.50	760.01	3.85
Spillover effect of tourism sector (SG: 2.35, TH 2.09)	S4	S4	0	0	193.990	55.635	2.44	0.33	254.78	1.20
	S4	S9	0	0	193.188	56.064	2.46	1.18	321.73	1.57
	S9	S4	0	0	192.202	58.075	8.35	0.50	760.01	3.85
Vaccinated and unvaccinated travelers permitted	S4	S9	0	0	152.067	74.497	10.11	5.02	956.70	6.71
	S4	S4	0	0	151.549	75.965	10.37	3.09	811.20	5.93
	S9	S4	0	0	148.717	78.057	26.66	1.97	1955.17	12.37
Medical cost of COVID cases (2×)	S4	S4	0	0	125.843	27.451	2.44	0.33	254.78	1.20
	S4	S9	0	0	125.038	27.814	2.46	1.18	321.73	1.57
	S9	S4	0	0	123.630	29.854	8.35	0.50	760.01	3.85
Percentage productivity loss due to quarantine (0%)	S4	S4	0	0	125.947	27.344	2.44	0.33	254.78	1.20
	S4	S9	0	0	125.143	27.801	2.46	1.18	321.73	1.57
	S9	S4	0	0	124.002	29.733	8.35	0.50	760.01	3.85
QALY loss saved from vaccination for symptomatic cases (0%)	S4	S4	0	0	125.938	27.379	2.44	0.33	254.78	1.20
	S4	S9	0	0	125.134	27.743	2.46	1.18	321.73	1.57
	S9	S4	0	0	123.962	29.758	8.35	0.50	760.01	3.85
CET (3 GDP per capita)	S4	S4	0	0	125.332	30.339	2.44	0.33	254.78	1.20
	S9	S4	0	0	121.904	31.768	8.35	0.50	760.01	3.85

Abbreviations: INB, incremental net benefit; SG, Singapore; TH, Thailand Testing strategy (when no quarantine): (S4) ART pre-departure and on arrival; (S9) no testing. Imported cases include (i) inbound travelers detected within the border and (ii) outbound travelers detected after return. Secondary cases include those infected by inbound travelers or returning outbound travelers.

## Data Availability

We declare that the data, models, and methodology used in the research are proprietary. Data are available on request to authors due to restrictions.

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
