# Peer review of "Economic Analysis of Border Control Policies during COVID-19 Pandemic: A Modelling Study to Inform Cross-Border Travel Policy between Singapore and Thailand"

_ijerph, 2023, doi:10.3390/ijerph20054011_

Round 1

Reviewer 1 Report

Thanks to the authors for sharing essential work on the economic analysis of border control policies during the COVID-19 pandemic. Overall, the manuscript was well written and presented sound modelling methods with clearly presented findings. The study raised an interesting and important question, as most existing studies haven't explored the economic impacts of border control policies on both the health and the non-health sector. Hence, a modelling study that included healthcare losses from COVID-19 transmissions due to cross-border travel policies would likely be appreciated. 

I think three are only minor issues that the authors can clarify or correct before the manuscript can be accepted for publication--only a few minor methodological errors and some text editing. 

First, all figures (Figures 1, 2, 3) were missing from the manuscript (PDF) or the supplement documents (MS Word) shared with the reviewers. Hence, I cannot evaluate if and how these figures help communicate the main messages of the study. This may be an editorial or managerial error, which should be corrected quickly.

Second, the sensitivity analysis is probably missing. While the model is logically sound, with three modules of the willingness-to-travel (WTT), the transmission model, and the economic model, all modules are interconnected, especially the transmission and economic models. For instance, the economic model's findings can be varied depending on the transmission model's assumptions. 

More specifically, since the timeframe of the model is October 2021, the transmission rate would not depend only on the vaccination effectiveness of the international travellers but also on the vaccination coverage of the host population in each traveller-receiving country, Singapore and Thailand. Therefore, the sensitivity analysis on how the vaccination rollout in each country can impact the study findings would be appreciated. A sensitivity analysis of other selected, crucial determinants of the willingness-to-travel (WTT), the transmission model, and the economic model can benefit the interpretation and generalization of the model, too. 

At the very least, suppose the authors somehow cannot provide the sensitivity analysis of some crucial parameters. In that case, the authors can discuss it as a limitation of the study, particularly its generalizability. 

Lastly, some minor aspects of the study presentation can be improved. For instance, on line 181, the authors explained that the economic model quantified the NMB of each country as the sum of the 13 components described in Table 2. But Table 2 presents the cost/benefit components from only Thailand's perspective. Maybe the authors rewrite the text to elaborate on how Table 2 demonstrates an example of the NMB components from Thailand's perspective. In that case, it'd be clear to the audience that another part of modelling also addresses the Singaporean's perspective.

Reviewer 2 Report

time range and a more specific definition of the purpose of the article are required in the abstract

for me personally, the literature review is missing - the introduction not providing enough literature references to set your research in a particular scientific context
sources referring to table 2 - is it personal opinions of authors or own reflection based on some literature? if the first option for me is very subjective and unproved

Figure 2 probably is missing from the text

Eq 5 and 6 are not correctly formatted into text

Conclusions can be broader discussed

Reviewer 3 Report

The analysis idea is interesting. However, it would be advisable to supplement the information on what testing/quarantine policy both countries ultimately chose, how it relates to the study results, and whether the policy was in line with the indications.

Round 2

Reviewer 2 Report

I accept text in such form